# Regulation of Mertk Surface Expression via ADAM17 and γ-Secretase Proteolytic Processing

**DOI:** 10.3390/ijms25084404

**Published:** 2024-04-17

**Authors:** Kevin C. Lahey, Christopher Varsanyi, Ziren Wang, Ahmed Aquib, Varsha Gadiyar, Alcina A. Rodrigues, Rachael Pulica, Samuel Desind, Viralkumar Davra, David C. Calianese, Dongfang Liu, Jong-Hyun Cho, Sergei V. Kotenko, Mariana S. De Lorenzo, Raymond B. Birge

**Affiliations:** 1Department of Microbiology, Biochemistry and Molecular Genetics, Center for Cell Signaling, Rutgers New Jersey Medical School, 205 South Orange Ave, Newark, NJ 07103, USA; kevinclahey@gmail.com (K.C.L.); christopher.varsanyi@rutgers.edu (C.V.); zw254@gsbs.rutgers.edu (Z.W.); fa453@gsbs.rutgers.edu (A.A.); ar2063@gsbs.rutgers.edu (A.A.R.); rp1099@rutgers.edu (R.P.); samuel.desind@rutgers.edu (S.D.); viraldavra@yahoo.com (V.D.); dc.calianese@gmail.com (D.C.C.); kotenkse@njms.rutgers.edu (S.V.K.); 2Department of Pathology, Immunology and Laboratory Medicine, Center for Immunity and Inflammation, Rutgers New Jersey Medical School, Newark, NJ 07101, USA; dongfang.liu@rutgers.edu (D.L.); jc2674@gsbs.rutgers.edu (J.-H.C.); 3Department of Cell Biology and Molecular Medicine, New Jersey Medical School, 185 South Orange Ave, Newark, NJ 07103, USA; delorems@njms.rutgers.edu

**Keywords:** Mertk, proteolysis, ADAM-17 ectodomain shedding, γ-secretase, Gas6, intracellular receptor trafficking, membrane expression

## Abstract

Mertk, a type I receptor tyrosine kinase and member of the TAM family of receptors, has important functions in promoting efferocytosis and resolving inflammation under physiological conditions. In recent years, Mertk has also been linked to pathophysiological roles in cancer, whereby, in several cancer types, including solid cancers and leukemia/lymphomas. Mertk contributes to oncogenic features of proliferation and cell survival as an oncogenic tyrosine kinase. In addition, Mertk expressed on macrophages, including tumor-associated macrophages, promotes immune evasion in cancer and is suggested to act akin to a myeloid checkpoint inhibitor that skews macrophages towards inhibitory phenotypes that suppress host T-cell anti-tumor immunity. In the present study, to better understand the post-translational regulation mechanisms controlling Mertk expression in monocytes/macrophages, we used a PMA-differentiated THP-1 cell model to interrogate the regulation of Mertk expression and developed a novel Mertk reporter cell line to study the intracellular trafficking of Mertk. We show that PMA treatment potently up-regulates Mertk as well as components of the ectodomain proteolytic processing platform ADAM17, whereas PMA differentially regulates the canonical Mertk ligands Gas6 and Pros1 (Gas6 is down-regulated and Pros1 is up-regulated). Under non-stimulated homeostatic conditions, Mertk in PMA-differentiated THP1 cells shows active constitutive proteolytic cleavage by the sequential activities of ADAM17 and the Presenilin/γ-secretase complex, indicating that Mertk is cleaved homeostatically by the combined sequential action of ADAM17 and γ-secretase, after which the cleaved intracellular fragment of Mertk is degraded in a proteasome-dependent mechanism. Using chimeric Flag-Mertk-EGFP-Myc reporter receptors, we confirm that inhibitors of γ-secretase and MG132, which inhibits the 26S proteasome, stabilize the intracellular fragment of Mertk without evidence of nuclear translocation. Finally, the treatment of cells with active γ-carboxylated Gas6, but not inactive Warfarin-treated non-γ-carboxylated Gas6, regulates a distinct proteolytic itinerary-involved receptor clearance and lysosomal proteolysis. Together, these results indicate that pleotropic and complex proteolytic activities regulate Mertk ectodomain cleavage as a homeostatic negative regulatory event to safeguard against the overactivation of Mertk.

## 1. Introduction

Mertk, a member of the TAM (Tyro-3, Axl, and Mertk) family, is a type I Receptor Tyrosine Kinase (RTK) with important functions in the regulation of inflammation and innate immunity [1,2]. Pioneering studies by the Lemke lab over the past two decades have described important homeostatic roles for Mertk and TAMs in maintaining tissue tolerance, immune suppression, and autoimmunity [3,4], whereby they exert complex and pleiotropic functions in many cell and tissue types that include cardiovascular tissue [5], microglia physiology, and neurodegeneration [6], blood-brain barrier regulation [7], as well as retinal biology and vision [8,9]. In genetic ablation models, mice lacking all three TAM receptors exhibit a systemic autoimmunity marked by highly activated antigen-presenting cells in vivo and ultimately develop an age-dependent systemic lupus erythematosus (SLE)-like pathology characterized by the production of antibodies to self-antigens such as anti-nuclear DNA antibodies and antibodies to histones [10]. Studies by Zizzo and Cohen also showed that Mertk is a critical apoptotic cell receptor expressed on M2c macrophages, whereby the loss of function of Mertk causes autoimmunity and inflammation [11]. In humans, Mertk has been linked by GWAS studies as a novel genetic risk for SLE and end-stage renal disease [12], and an increase in the soluble ectodomain of Mertk has been observed in SLE [13], suggesting translational relevance to the mouse studies in the relationship to the immunopathogenesis of SLE. Mechanistically, on macrophages and DCs, TAM receptors act as inhibitory receptors that dampen inflammation, suppress inflammatory cytokine production [14,15], and enhance the production of pro-resolving mediators, including prostaglandins and lipoxins [12].

Mertk is also a critical receptor for efferocytosis and the homeostatic clearance of dying cells, a process that is critical for the maintenance of immunological tolerance and the resolution of inflammation [13]. The role of Mertk in regulating efferocytosis is prominent among the TAM family in macrophage subsets, as macrophages with a single knockout of Mertk show a defective clearance of apoptotic cells [11,13,14,15,16,17]. The regulation of efferocytosis by Mertk appears to be binary in nature, whereby efficient apoptotic cell uptake and digestion requires a tethering co-receptor, most notably MFG-E8/αvβ5(3) integrin [18,19,20] or TIM-4 [14,21], an idea sometimes called the “tethering and tickling” effect [16,22,23]. Upon ligand stimulation, Mertk is activated by autophosphorylation, whereby main endogenous ligands’ Protein S (PROS1) or Growth arrest-specific 6 (GAS6) ligands bind to and opsonize phosphatidylserine (PtdSer) exposed on the surface of apoptotic cells to bridge Mertk [24,25,26]. Ultimately, Mertk-mediated efferocytosis results in the trafficking of the apoptotic cargo to a phagolysosomal itinerary for the production of tolerogenic cytokines such as IL-10 and TGF-β, resulting in the maintenance of tolerance [27].

More recently, studies have shown that Mertk is expressed on tumor-associated macrophages and has a vital role in the clearance of dying tumor cells (and tumor antigens) and the subsequent maintenance of the immune suppressive pro-tumor microenvironment [28,29,30,31,32,33]. Such preclinical studies also demonstrated that inhibiting Mertk may have therapeutic value in oncology as either a single agent or in combination with other therapies, such as immune checkpoint inhibitors [29,33]. In this capacity, antibodies to Mertk or small molecular tyrosine kinase inhibitors that target Mertk can synergize with anti-PD1 therapeutics to stimulate host anti-tumor immune activation, including the activation of host anti-tumor antigen-specific T cells [29,33]. Importantly, at the mechanistic level, both genetic knockout studies and pharmacological studies indicate that Mertk inhibition (as an anticancer immune-oncology effect) appears to involve the regulation of efferocytosis as a key mechanism [17,29,33]. These studies also suggest that the priori regulation of Mertk expression may have important consequences for Merk’s biological activity and functional outcomes. Indeed, consistent with this interpretation, TAM receptors, including Mertk, while not commonly mutated in cancers [34], can often become up-regulated by enhanced transcriptional activation [35] or by post-translational effector mechanisms such as receptor ectodomain shedding and ubiquitin-mediated degradation [12,36]. In addition, Mertk on macrophages and tumor-associated M2 macrophages can be up-regulated transcriptionally [15]. On tumor-associated macrophages, Mertk is up-regulated and associated with cold tumors by enhancing efferocytosis in actively proliferating solid cancers and by inducing tolerogenic cytokines in the tumor microenvironment [37].

Mertk is also regulated at the post-translational level by ectodomain shedding, a proteolytic function that produces a decoy receptor incapable of Gas6- and Pros1-induced receptor activation [38]. The proteolytic cleavage of Mertk is mediated by several enzymes, including members of the ADAM (a disintegrin and metalloproteinase) family of proteases. ADAM17, also known as TACE (TNF-α converting enzyme), is the most well-characterized enzyme involved in Mertk cleavage [39]. ADAM17 is a transmembrane protease that cleaves a variety of substrates, including cytokines, growth factors, and cell surface receptors. ADAM17 cleaves Mertk at a site within the extracellular domain, releasing a soluble form of the receptor (sMertk) [39]. In addition, although less well studied, Mertk and other TAMs have also been reported to become processed by an intracellular γ-Secretase-mediated pathway, resulting in the generation of an intracellular fragment with a cytosolic trafficking itinerary [38].

Given the importance of Mertk’s expression, efferocytosis, and immune outcomes, here we investigated the post-receptor and post-translational processing of Mertk, both homeostatically and under conditions of Gas6-mediated receptor activation and signaling. Towards this goal, we generated a novel reporter line comprised of a Flag-Mertk extracellular domain and a Mertk-EGFP-Myc intracellular domain, with the latter used to interrogate the potential trafficking of an intracellular fragment. Our results show that Mertk is homeostatically processed and cleaved by an ADAM17 and γ–Secretase combination duet. Moreover, we observed that following receptor activation by Gas6, Mertk appears to be targeted in a phagolysosomal degradative itinerary. Neither under homeostatic nor by ligand-inducible Mertk activation did we observe a stable accumulation of an intact intracellular Mertk fragment. These data suggest that the regulated proteolysis of Mertk is tightly controlled to regulate surface receptor levels, potentially to safeguard against overexpression negatively.

## 2. Results

### 2.1. The Differentiation of THP-1 Cells by PMA Promotes Robust Mertk Up-Regulation at the Transcriptional and Translational Level

When the human leukemia monocytic cell line THP-1 was treated with 100 ng/mL of phorbol 12-myristate 13-acetate (PMA) for up to 48 h to activate Protein Kinase C (PKC) and differentiate monocytes into macrophages, Mertk was observed to be robustly up-regulated between 12 and 24 h, reaching a maximum at 24 h, consistent with previous observations that Mertk is up-regulated in differentiated macrophages (Figure 1A, top panel) [11,15]. To compare Mertk’s up-regulation with the repertoire of other TAM receptors (Tyro-3 and Axl) and endogenous TAM ligands (Gas6 and Protein S), replicate detergent lysates from the PMA-treated THP-1 cells were subsequently probed with anti-Axl, anti-Tyro-3, anti-Gas6, or anti-Pros1 antibodies (Figure 1A, lower panels for TAM receptors and Figure 1B, for TAM ligands). H1299 (human lung epithelial-like adenocarcinoma line) cells, Hela (human cervical carcinoma cell line) cells, and BEAS2B (immortalized non-tumorigenic lung cancer cell line) cells were used as controls for antibody reactivity for Mertk, Axl, and Tyro3, respectively, based on previous observations in our laboratory. As indicated in the lower panels, while Axl was not expressed or induced in PMA-differentiated THP-1 cells, Tyro-3 was constitutively expressed. The static expression patterns of Axl (absent) and Tyro3 (constitutive) are clearly in stark contrast to the dynamic regulation of Mertk in PMA-treated THP-1 cells.

With respect to Gas6 and Pros1 (the two predominant endogenous ligands for Mertk), Gas6 showed a reciprocal expression pattern to Mertk (shown are representative of several independent experiments), whereby Gas6 expression was observed up to 12 h, after which its expression was strongly down-regulated in a time-frame coincident with Mertk up-regulation (Figure 1B). By contrast, Pros1 was induced in PMA-differentiated THP-1 cells (those shown are representative of several independent experiments), more reminiscent of Mertk up-regulation. Protein quantification is shown in Figure 1C,D. The qRT-PCR of 72-h post-PMA-treated THP1 cells showed similar expression patterns, most notably a robust reciprocal regulation of Gas6 and Mertk at the mRNA level (Figure 1E). This reciprocal pattern of expressions of Gas6 and Mertk defines a non-autocrine pattern of the expression of Gas6 in PMA-differentiated Mertk-positiveTHP1 cells, for example, as depicted by a “ying-yang” inverse reciprocal relationship (Figure 1F).

### 2.2. Homeostatic Regulation of Endogenous Mertk Cleavage by ADAM17 and γ-Secretase

To further characterize the up-regulation and processing of Mertk following PMA administration, we compared Mertk immunostainings with two independent Mertk mAbs, which include an N-terminal mAb that recognizes the ectodomain of Mertk and a C-terminal mAb that recognizes the intracellular cytoplasmic domain of Mertk. As shown in Figure 2A (top panel) and consistent with results in Figure 1, PMA-inducible Mertk reproducibly showed the characteristic doublet band employing an N-terminal ectodomain mAb, denoting a fully glycosylated and partially glycosylated form of full-length Mertk (180 kDa and 140 kDa, respectively) between 12 and 24 h following PMA administration and remained stable for up to 120 h. When replicate lysates were probed with the Mertk mAb that recognizes the intracellular domain of Mertk (second panel), Mertk was noted again to be prominently up-regulated between 12 and 24 h typical of the pattern of PMA-inducible Mertk expression. In contrast, while the N-terminal mAb detected mainly the 180 kD fully glycosylated form, the C-terminal mAb mainly detected a 140 kD partially glycosylated form, as well as a 90 kD fragment that peaked distinctly ~48 h following PMA treatment. Indeed, high ratios of 180 kDa/140 kDa bands using anti-N-terminal Mertk and low ratios of 180 kDa/140 kDa bands using anti-C terminal Mertk suggest both continuous processing through the endoplasmic-reticulum/trans-Golgi network as well as steady state ectodomain protease cleavage (Figure 2A,D). The appearance of a de novo-generated 90 kDa band post 48 h also likely indicates intracellular proteolysis or a feedback mechanism stabilizing an under-glycosylated form of Mertk that is pre-empted by the membrane localization of a more mature form of Mertk.

The aforementioned difference in Mertk isoforms detected by N- versus C-directed antibodies predicts the basal constitutive proteolytic cleavage of Mertk under homeostatic conditions. Previous studies have reported that Mertk can be processed by two discrete post-translational proteolytic events. The first involves an ADAM17-mediated ectodomain shedding [38,39], producing a shed ectodomain decoy receptor and a membrane-proximal C-terminal fragment (mb-cMertk). Subsequently, but less well-studied than ADAM17, the Mertk mb-cMertk fragment may undergo a sequential proteolytic event mediated by γ-secretase to produce a membrane-released intracellular soluble fragment (sMertk) [40]. Since cleavage by γ-secretase occurs adjacent to the membrane domain and is predicted to result in a fragment without the anchored membrane domain, cytoplasmic-cMertk (cyto-cMertk) has been proposed to traffic with an intracellular itinerary, including reports of a cytosolic-to-nuclear translocation in some cells’ systems (Borgman, bioRxiv, 2020) [41].

To assess whether THP-1 cells express requisite proteases for Mertk cleavage and whether PMA concomitantly induces proteases, we examined ADAM17 expression as well as the expression of components of the γ-secretase complex, including Presenilin 1 (Pres1), Presenilin 2 (Pres2), and Niscastin [42] by Western blotting (Figure 2A, lower panels). As indicated, PMA induced the expression of ADAM17 (peaking at 6–12 h before the peak expression of Mertk), and expression remained stable thereafter. THP-1 cells show the constitutive expression of components of the γ-secretase complex, although Pres2 showed a modest induction peaking at 6–12 h (Figure 2A, lower four panels, respectively) in a time frame similar to that of ADAM-17. These data indicate that upon Mertk’s up-regulation by PMA, all ADAM17/γ-Secretase proteolytic cascade components are concomitantly expressed in THP-1 cells.

To examine whether Mertk is proteolytically processed under homeostatic conditions, we used a series of protease inhibitors that include (i) GW280264X (GW), an ADAM17 inhibitor [43], (ii) DAPT, a γ-secretase inhibitor [44], and (iii) MG132, a proteasome inhibitor [45]. As shown in Figure 2B, lane 2, the treatment of GW280264X (GW) stabilized the full-length Mertk as evident by the enhanced detection of the mature form of Merk. Notably, under these conditions, we did not detect a cytoplasmic cleavage species of Mertk (see arrow in the figure). Hence, to address whether the cytoplasmic species was labile to constitutive proteolytic degradation, we treated PMA-differentiated THP-1 cells with either DAPT or MG132 (lanes 3 and 4). Interestingly, a notable ~75 kD cytosolic fragment was identified under these conditions, which was largely blocked by pretreatment with the ADAM17 inhibitor GW280264X (lanes 5 and 6). Moreover, the ~75 kD migrated slightly faster in the MG132-treated cells, suggesting that upon ADAM17’s cleavage of Mertk, there was a rapid and sequential dual protease event, with the first protease being mediated by the γ-secretase complex yielding a membrane-bound fragment (mb-cMertk fragment), and the second being mediated by generalized ubiquitin-mediated proteasome degradation to eliminate the cytosolic fragment. Pretreatment with GW280264X strongly abrogated the stabilization effects of both DAPT and MG132, suggesting that the ectodomain shedding and the emergence of the mb-cyto-cMertk are requisite for the subsequent stabilization of truncated forms of Mertk by DAPT or MG132 (Figure 2B, lanes 5 and 6). Consistent with this idea, when comparing the soluble released N-terminal fragment of Mertk (sMertk) in the presence or absence of DAPT (Figure 2C), unlike the cMertk fragment which requires DAPT stabilization (lane 4), in contrast, sMertk, an upstream cleavage event of ADAM17, is independent on DAPT whereby control produced equivalent amounts of sMertk in the media. This suggests that the product of ADAM17 cleavage (mb-cMertk) is a necessary requirement for subsequent γ-secretase cleavage (Figure 2C, compare lanes 4 and 5). A model for the sequential processing of Mertk by ADAM17 and γ-secretase is depicted in Figure 2D.

To examine whether Mertk is proteolytically processed under homeostatic conditions, we used a series of protease inhibitors that include (i) GW280264X (GW), an ADAM17 inhibitor [43], (ii) DAPT, a γ-secretase inhibitor [44], and (iii) MG132, a proteasome inhibitor [45]. As shown in Figure 2B, lane 2, the treatment of GW280264X (GW) stabilized the full-length Mertk as evident by enhanced detection of the mature form of Merk. Notably, under these conditions, we did not detect a cytoplasmic cleavage species of Mertk (see arrow in the figure). Hence, to address whether the cytoplasmic species was labile to constitutive proteolytic degradation, we treated PMA-differentiated THP-1 cells with either DAPT or MG132 (lanes 3 and 4). Interestingly, a notable ~75 kD cytosolic fragment was identified under these conditions, which was largely blocked by pretreatment with the ADAM17 inhibitor GW280264X (lanes 5 and 6). Moreover, the ~75 kD migrated slightly faster in the MG132 treated cells, suggesting that upon ADAM17’s cleavage of Mertk, there was a rapid and sequential dual protease event, with the first protease being mediated by the γ-secretase complex yielding a membrane-bound fragment (mb-cMertk fragment), and the second being mediated by generalized ubiquitin-mediated proteasome degradation to eliminate the cytosolic fragment. Pretreatment with GW280264X strongly abrogated the stabilization effects of both DAPT and MG132, suggesting that the ectodomain shedding and the emergence of the mb-cyto-cMertk is requisite for the subsequent stabilization of truncated forms of Mertk by DAPT or MG132 (Figure 2B, lanes 5 and 6). Consistent with this idea, when comparing the soluble released N-terminal fragment of Mertk (sMertk) in the presence or absence of DAPT (Figure 2C), unlike the cMertk fragment which requires DAPT stabilization (lane 4), in contrast, sMertk, an upstream cleavage event of ADAM17, is independent on DAPT whereby control produced equivalent amounts of sMertk in the media. This suggests that the product of ADAM17 cleavage (mb-cMertk) is a necessary requirement for subsequent γ-secretase cleavage (Figure 2C, compare lanes 4 versus 5). A model for the sequential processing of Mertk by ADAM17 and γ-secretase is depicted in Figure 2D.

### 2.3. Ligand-Dependent Activation of Endogenous Mertk and MTag upon Stimulation with Recombinant γ-Carboxylated Gas6

The abovementioned observations that Mertk is constitutively and sequentially regulated by ADAM17 ectodomain cleavage and γ-secretase cytosolic cleavage under homeostatic conditions suggests steady-state Mertk regulation to maintain basal receptor expression levels. To assess whether ligand-inducible Mertk activation alters homeostatic processing, and in particular, whether the activation of Mertk and tyrosine phosphorylation stabilizes the intracellular cytosolic fragment, possibly akin to a γ-secretase biologic inhibitor mimetic or an MG132 biologic inhibitor biological mimetic, we tested recombinant active Gas6 (γ-carboxylated) and recombinant inactive Gas6 (Warfarin-treated and non-γ carboxylated) cloned and expressed in-house (strategy shown in Figure 3A) [46,47]. As shown in Figure 3B, the addition of Vitamin K to the culture media of Gas6-expressing HEK cells produces the secretion of a Gas6 protein that is robustly γ-carboxylated (Gas6-K), whereas the culture of the Gas6-expressing HEK cells with Warfarin (an inhibitor of VCOR1) produces the secretion of a completely non-γ-carboxylated Gas6 protein (Gas6-W) (compared anti-Gas6 versus anti-Gla Western blots are in Figure 3B). However, when PMA-differentiated THP-1 cells were treated with 10 nM of the aforementioned Gas6 proteins, only the Gas6-VK, but not the Gas6-W, induced prominent Mertk tyrosine phosphorylation, a hallmark of Mertk activation and post-receptor signaling (see also Figure 5C). These data support our previous observations that γ-carboxylated (via subsequent binding to PS+ particles) is necessary for Gas6 to be active as a Mertk ligand and induce Mertk tyrosine phosphorylation and signaling.

By interrogating both biologically active (Gas6 Vitamin K) versus biologically inactive (Gas6 Warfarin) species of Mertk, we next tested whether active Gas6 could phenocopy effects of γ-secretase or proteasome inhibitors and stabilize the intracellular Mertk fragment (Figure 3C). PMA-differentiated THP-1 cells were treated with Gas6-VK or Gas6-W for 30 min, after which detergent lysates were monitored for both full-length and proteolytic processed receptors (Figure 3C, arrow). Interestingly, neither active Gas6-VK nor inactive Gas6-W stabilized a Mertk intracellular fragment as a proteolytic mimetic under the conditions utilized in the assay. However, in the lanes from cells treated with the Gas6-VK treatment, while similar levels of endogenous sMertk levels were observed to be shed into the supernatant (devoid of the ADAM inhibitor samples, which blocks ectodomain cleavage), intracellular levels of Mertk decreased when treated with active Gas6-VK compared to a mock control or inactive Gas6-W (a 45% reduction in sMertk vs. the mock, while the Gas6-W resulted in only an 11% reduction vs. the mock) (Figure 3C, top panel, Figure 3D). These data suggest that active Gas6-VK does not phenocopy inhibitors to stabilize a stable intracellular fragment, but rather that Gas6-VK likely destabilizes or degrades the full-length receptor.

### 2.4. Stable Ectopic Expression of Flag-Mertk-EGFP-Myc Chimeric Reporter Construct in THP1 Cells

Given the potential for Mertk to be intracellularly cleaved and release cytosolic bound forms of Mertk, we generated a novel chimeric Mertk reporter line with the capacity to monitor the membrane localization and trafficking of Mertk using fluorescent microscopy. Towards this goal, we engineered, cloned, and ectopically expressed a Mertk reporter construct that contains an N-terminal FLAG-tag fused in frame to human Mertk with tandem C-terminal EGFP and Myc tags (called MTag). (Figure 4A–C). When ectopically expressed in THP1 cells and single-cell clones selected and passaged (Figure 4B), the Flag-Mertk-EGFP-Myc construct displayed a typical expression pattern for processed Mertk as evident by anti-Flag (N-terminal) immunostaining and anti-Myc (C-terminal immunostaining) as well as single collapsible bands when cells were treated with Tunicamycin (an inhibitor of N-linked oligosaccharide glycosylation [48]) or Brefeldin (an inhibitor of protein trafficking from the Endoplasmic Reticulum (ER) to the Golgi and O-linked glycosylation [49]), suggesting the proper trafficking and transport of the chimeric receptor (Figure 4C); note that due to the 27 kDa EGFP domain, the larger and smaller glycoforms appear at 250 kDa and 200 kDa. When MTag-expressing cells were gated as EGFP/Flag (lower left panel) or EGFP/N-Mertk ectodomain (lower right), single, double-positive peaks were observed (Figure 4B), confirming the surface expression of the tagged construct via flow cytometry.

To assess whether the MTag construct behaved analogous to WT (as in the PMA-differentiated THP1 cells in see Figure 2B), MTag-expressing cells were treated with (i) GW280264X, the ADAM17 inhibitor (Figure 4D, lane 2), (ii) DAPT, the γ-secretase inhibitor (lane 3), and (iii) MG132, the proteasome inhibitor (lane 4). Analogous to WT Mertk, the sequential nature of MTag cleavage was confirmed using GW280264X, whereby ectodomain fragment stabilization in the conditioned media was completely blocked by GW280264X but not in the DAPT or MG132 treatment groups, again confirming that ADAM17 cleavage is necessary for γ-secretase cleavage to occur using a second reporter Mertk construct (lane 2). In contrast, the C-terminal fragment, containing the Mertk intracellular domain and the EGFP-Myc duet, was only stabilized when DAPT or MG132 was used, analogous to the results obtained with WT Mertk in the PMA-differentiated cells. (Figure 4D, last two lanes).

### 2.5. Gas6-Mediated Activation of Mertk Drives Mertk to a Distinct Proteolytic Itinerary

To assess the fate of Mertk following ligand stimulation, we treated MTag Mertk with recombinant Vitamin K Gas6 or Warfarin Gas6 proteins, respectively (Figure 5). As indicated in Figure 5A, when recombinant Gas6 was used to treat cells, only the active Gas6-VK, but not inactive Gas6-W, resulted in the destabilization of the Flag-PE signals (flow panel C). Cells were pretreated with cycloheximide (CHX) to ensure we only assessed surface-bound Mertk and not de novo resynthesized Mertk. The quantification of surface Flag-PE is shown in Figure 5B, and the activation/tyrosine phosphorylation of Mertk is shown in Figure 5C. Figure 5D shows a time-course of Gas6-VK (active)-treated Mtag cells, showing a time-dependent receptor down-regulation by Western blotting using three distinct antibodies (the N-terminal region of the construct (noted as FLAG and N-Mertk) and the C-terminal region of the construct (noted as Myc)).

Previous observations reported that Mertk or C-terminal Merk could traffic into the nucleus, possibly with functional consequences for binding chromatin (Borgman, bioRxiv 2020). To investigate the subcellular localization of MTag Mertk, we employed confocal microscopy, as described in Figure 6. As indicated, WT Mertk-EGFP showed clear plasma membrane localization, while DAPT pr MG132-treated cells show a hybrid plasma membrane cytosolic localization (compare Figure 6A, compare Figure 6B,C). Following either DAPT or MG132 pretreatment, we did not observe the nuclear co-localization of cytosolic Mertk-EGFP via confocal microscopy but rather a punctate cytosolic pattern with DAPT and a diffuse cytosolic pattern with MG132. In Figure 6B, and quantified in Figure 6C, the effects of the active (Gas6-Vitamin K) and inactive (Gas6-Warfarin) are shown whereby active Gas6, but not inactive Gas6, drives receptor down-regulation from the cell surface, suggesting intracellular trafficking to a degradative itinerary. Models for homeostatic and ligand-inducible proteolysis are also depicted in the cartoon in Figure 6.

### 2.6. Mertk K619M Kinase-Dead Mertk Retains Homeostatic Processing by ADAM17 and γ-Secretase

As the previous data with Gas6-VK leading to Mertk degradation suggests a role of activation or phosphorylation, we created a mutant Mertk incapable of autophosphorylation in which the methionine in the ATP-binding site (K619M) was substituted in the tagged construct. The K619M substitution, which mutates the coordinating lysine of the ATP-binding pocket, prevents downstream autophosphorylation (Figure 7A,B). Kinase dead Mertk was validated for inactivation (Figure 7B) using 10 nM of Gas6-VK, and the cleavage patterns shown in Figure 6C indicate that any effects observed in the K619M mutant are independent from phosphorylation. Indeed, when K619M Mertk was treated in the presence of GW280264X, DAPT, and MG132, the K619M mutant Mertk showed similar cleavage patterns as WT Mertk when treated with the DAPT and MG132 inhibitors as both stabilized C-terminal fragments (Figure 7C,D (right panels)). Collectively, these data suggest a two-tiered level of Mertk proteolytic down-regulation, the first tier homeostatically involving sequential proteolytic cleavages by ADAM17 and γ-secretase, and the second tier involving Gas6-VK-mediated trafficking to the endolysosome for intracellular degradation. Such activating of independent sequential activation by ADAM17 and γ-secretase differs from classical Notch signaling, whereby Notch ligands trigger proteolytic processing and the stabilization of the Notch intracellular domain (Figure 7E).

## 3. Discussion

Employing both PMA-differentiated human THP-1 cells, as well as an engineered human Flag-Mertk-EGFP-Myc reporter construct that expresses a chimeric Mertk recombinant fusion protein, we show that two distinct patterns of proteolysis events post-translationally regulate Mertk. Under basal homeostatic conditions, i.e., without ligand stimulation, Mertk is regulated by a sequential regulatory cleavage event, with the first portion involving ectodomain cleavage by ADAM10/ADAM17 and the second portion begin a tandem event involving a transmembrane/intracellular cleavage by γ-secretase to produce a cytosolic intracellular cleaved fragment. Additionally, following Gas6 ligand-induced activation, Mertk is targeted to a lysosomal degradation itinerary by receptor-mediated endocytosis. Both surface ectodomain shedding and the γ-secretase release of the cytosolic fragment, as well as Gas6-mediated endo-lysosomal trafficking, promote a common Mertk-negative regulation. Together, these studies indicate that Mertk receptor levels are tightly regulated by distinct ligand-dependent homeostatic mechanisms as well as ligand-independent mechanisms.

The homeostatic process of Mertk post-translational processing, first involving ADAM17-mediated ectodomain shedding, releases a stable ectodomain fragment that can bind and sponge exogenous ligands Gas6, Pros1 and sequester them in decoy dead-end complexes separated from functional trans-membrane-signaling receptors. In the present study, we observe the stable ectodomain fragment from both native PMA-treated THP-1 cells as well as those when Mertk is ectopically expressed as a chimeric reporter line. In both expression systems, ectodomain shedding is fully blocked using an ADAM17 inhibitor. These data are consistent with previous in vivo data showing circulating soluble ectodomains of Mertk [39] and that the elevated levels of soluble Mertk that are noted in vivo undergo pathophysiological conditions that include systemic lupus erythematosus (SLE) [15,50], atherosclerosis [51], SLE-induced kidney disease [52], an acute inflammatory response [36], and a COVID-19 cytokine storm [53]. Collectively, these data indicate that Mertk ectodomain-shedding functions likely offset the tolerogenic effects of full-length Mertk signaling.

By contrast to ADAM17-mediated ectodomain shedding and its consequence as a functional decoy to bind ligands (Gas6, Pros1, and possibly others), the physiological and/or non-physiological functions of the intracellular cytosolic fragments appear to be more elusive. Using both PMA-treated THP-1 cells and the chimeric reporter lines noted above; we observe that the ADAM17-cleaved and membrane-proximal C-terminal fragment (mb-cMertk) is rapidly processed by the γ-secretase complex to produce a cytosolic Mertk fragment. Evidence for this process is supported by the clear and robust stabilization of mb-cMertk by both the γ-secretase inhibitor (264XGW280) and the proteosome inhibitor (MG132). Together, this indicates a rapid γ-secretase/proteasome combination on the ectodomain-cleaved Mertk receptor to incapacitate the potential for the intracellular signaling of a cytosolic form of Mertk. Using the chimeric Flag-Mertk-EGFP-Myc construct, which was, in part, designed and developed to monitor an intracellular trafficking itinerary for Mertk (cMertk-EGFP), we observed distinct intracellular staining patterns between mb-cMertk (γ-secretase inhibitors stabilized a granular and vesiculated staining pattern) and cMertk (proteasome inhibitors stabilized a cytosolic staining pattern). While some previous reports have indicated the nuclear localization of full-length or processed Mertk under the homeostatic conditions described here, we did not observe notable Mertk-EGFP localization in the nucleus.

The observation mentioned above that the ligand-independent homeostatic regulation of Mertk produces a functional ectodomain trap (decoy receptor) and a γ-secretase/proteosome degraded non-functional intracellular fragment implies a negative regulation of Mertk at steady-state. To address the fate of WT Mertk (PMA-treated THP-1 cells) or the chimeric Mertk chimeric reporter following Gas6/ligand-inducible Mertk activation, we stimulated Mertk expressing cell lines with active Gas6 (produced in Vitamin K-treated producer cells) versus inactive Gas6 (produced in Warfarin-treated producer cells) as previously reported. In contrast to the homeostatic regulation, we observed reduced ADAM17-mediated ectodomain shedding but rather reduced total Mertk levels via the trafficking via endo-lysosomal trafficking. Moreover, we did not observe the ligand-dependent stabilization of the Mertk intracellular domain (mb-cMertk or cMertk), implying that under the conditions of Gas6 activated and employed in this study, Gas6-mediated Mertk activation did not serve as a “γ-secretase inhibitor mimetic” or a “proteasome inhibitor mimetic” to stabilize the Mertk intracellular fragment.

Classic examples of the ligand-inducible stabilization of an intracellular receptor fragment include, for example, the Wnt/Frizzled/LRP4-mediated stabilization of β-catenin [54] and the Notch ligand (Delta)-Notch-mediated stabilization of the Notch intracellular domain (NICD) [55,56]. However, under the conditions employed in this study, we did not find evidence of mb-sMertk or sMertk stabilization after the activation of Mertk by Gas6. Whether some additional co-stimulatory factors might be required to alter the intracellular processing of Mertk to stabilize mb-cMertk or cMertk remains to be determined. For example, more recent studies indicate that Mertk can be activated by non-canonical TUBBY/TUB-like ligands [57,58]. Additionally, Mertk signaling is enhanced when Gas6 or Pros1 is associated with a source of PS, such as apoptotic cells, PS-positive stressed cells or exosomes [46,47]. Whether Gas6-PS lipid hybrids alter the activation of Mertk or stabilize the sMertk intracellular fragment, or whether that cell type specifically occurs, should be further investigated.

The present study indicates complex and multifactorial post-translational proteolytic events to negatively regulate Mertk cell surface expression. Several previous studies using Mertk KO mouse models have shown strong phenotypes for loss-of-function Mertk, including an (i) enhanced susceptibility to endotoxin [59], (ii) failed efferocytosis and polarization of macrophages [13], (iii) susceptibility to chronic inflammation and SLE-like autoimmunity [11,15], and (iv) an auto-antibody production and rheumatoid arthritis [60,61]. Moreover, as noted above, increased circulating levels of soluble Mertk ectodomain are associated with chronic inflammatory conditions, suggesting that the pathological or controlled regulation of ADAM17 ectodomain shedding might phenocopy the loss of Merk functional biology to counteract tolerogenic Merk functions [39]. In contrast, experimental strategies that block ADAM17 ectodomain shedding, particularly strategies to knock-in ADAM17 mutant Mertk in mice, appear to have a gain-of-function action. ADAM17 cleavage-resistant mice show gain-of-function activity in NASH, leading to pathological fibrosis [12,62] and protective therapeutic values in clearing atherosclerotic plaques [39,51], suggesting complex phenotypic outcomes. Nevertheless, these latter observations suggest that ADAM17-mediated ectodomain cleavage is potentially targetable, either to increase shedding and reduce tolerance or decrease shedding and enhance immunogenic signaling. Given the emerging importance of Mertk in immuno-oncology and ICI, the role of ADAM17 cleavage-resistant Mertk and targeting ADAM17 as a therapeutic strategy deserves further investigation.

## 4. Materials and Methods

### 4.1. Design of a Flag-hMertk-EGFP-Myc Chimeric Plasmid Construct

To create the multi-tagged Mertk construct, the mature peptide sequence of Mertk (NM_006343.2) was inserted into a pMSCVpuro backbone vector. Preceding the Mertk insert is an interferon–γ receptor 2 signal peptide (NM_005534.3) for surface localization, a FLAG tag sequence, and a linker sequence containing a Sal1 restriction site. An AvrII restriction site was inserted within the transmembrane domain that resulted in a Phe→Leu change in one residue. After the C-terminus of Mertk, a linker containing a PacI restriction site was added, followed by an Emerald GFP motif, another linker, and finally, a Myc tag motif. Restriction sites at the ends of the construct, an N-terminal XhoI and a C-terminal EcoRI were also inserted for future cloning purposes. The final construct was transfected into EPI400 bacteria. For the Mertk kinase dead mutant, the K619M mutation within the ATP-binding site, site-directed mutagenesis was separately performed on the initial construct. 

### 4.2. In vitro Cell Culture

THP-1 cells were cultured with 5% (*v*/*v*) heat-inactivated FBS (R&D Systems, Flowery Branch, GA, USA) in RPMI-1640 (Mediatech Inc., Manassa, VA, USA)and were maintained at a density of less than 5 × 10^6^ cells/mL to ensure >90% viability. All cells were kept in a 37 °C humidified incubator with 5% CO_2_ with passage numbers below 40. 4.3. Quantitative Real-Time PCR and Sequences of Primers (Table 1).

Total RNA was isolated from a 5x10^6^ cell pellet of THP-1 cells, treated with or without PMA for 72 h, using the Qiagen RNeasy Mini Kit (74104, Qiagen, Hilden, Germany) following the manufacturer’s protocol. cDNA was generated using the High-Capacity cDNA Reverse Transcription Kit (4368814, Applied Biosystems, MA, USA).

Subsequently, a Quantitative Real-Time PCR was conducted on the Bio-Rad CFX96 Touch Real-Time PCR Detection System (Bio-Rad Laboratories, CA, USA). The PCR reactions were set up in a 96-well plate with approximately 100 ng of cDNA per reaction. Each reaction had a total volume of 20 μL, comprising 2 μL of cDNA, 10 μL of iTaq SYBR Green Supermix (Bio-Rad Laboratories, CA, USA), 6 μL of water, 0.5 μM of forward primer, and 0.5 μM of reverse primer. Cycling conditions included an initial denaturation at 94 °C for 2 min, followed by 40 cycles of denaturation at 95 °C for 15 s and annealing/extension at 60 °C for 1 min. Triplicate reactions were performed for each gene, including no-template controls (water instead of cDNA). The relative expression levels of Tyro, Axl, MerTK, Gas6, and Protein S mRNA were determined using the 2^(−ΔΔCT)^ method, normalized to β-Actin.

### 4.3. Transduction of THP-1 Cells with the Tagged Mertk Construct

THP-1 cells were retrovirally transduced with viral particles created after transfecting PLAT-A cells using LipoD293 (SignaGen SL100668, Ijamsville, MD, USA) and a pMSCVpuro plasmid containing the Mertk construct. Tagged Mertk constructs were created by adding a FLAG-tag sequence after the Mertk signal peptide, while an EGFP motif and a Myc tag were added after the C-terminal end. Viral particles were concentrated and were added to THP-1 cells for transduction. After 72 h, transduced THP-1 cells were kept under selection media (2 µg/mL of puromycin) for two weeks and then sorted into single cells based on GFP expression. Clones were selected based on proliferation rates and assessed via flow cytometry and Western blotting to validate the expression of the construct.

### 4.4. Monocytic THP-1 cell Differentiation to Macrophages

THP-1 cell differentiation to macrophages was conducted using 100ng/mL of phorbol-myristate acetate (PMA; Sigma-Aldrich, P1585-1MG, Darmstadt, Germany) for 72 h. Cells were plated at 2 × 10^6^ cells in 3 mL of RPMI-1640 with 5% (*v*/*v*) heat-inactivated FBS per well of a 6-well plate. After 48 h of differentiation, the media was refreshed.

### 4.5. C-Terminal Fragment Stabilization

C-terminal fragments of Mertk were stabilized with either 5 μM DAPT (D5942-5MG, Sigma Aldrich, MO, USA) or 10 μM MG132 (S2619, Selleckchem, TX, USA) in serum-free RPMI media for 4 h. To inhibit Mertk’s cleavage by ADAM17, 3 μM GW280264X (MedChemExpress, NJ, USA), an ADAM10/17 inhibitor, was used either alone or in combination with DAPT or MG132. After 4 h, cells were lifted via pipetting and centrifuged at 300× *g* for 10 min. Cell media was harvested for the analysis of sMertk and the cell pellet was harvested for fragment analysis.

### 4.6. Western Blotting

For cell lysates, cells were lysed with a RIPA buffer (R0278-50ML, Sigma Aldrich, Burlington, MA, USA) and a Halt 100× protease/phosphatase inhibitor cocktail (1861281, Thermo Scientific, Rockford, IL, USA). After centrifugation, a final protein concentration of 2–8 mg/mL was calculated using the DC Protein Assay (500-0116, Bio-Rad, Hercules, CA, USA). For the collection and analysis of soluble Mertk in the cell culture media, media were concentrated using a 10 kDa molecular weight cut-off centrifugal filter (UFC501096, Amicon, Darmstadt, Germany). For both the cell lysate and concentrated media, a 6 X SDS protein-loading buffer was prepared (341 mM of Tris-HCl, 8.2% SDS, 45.5% glycerol, 0.03% bromophenol blue, 9% β-mercaptoethanol) and added to the prepared protein. Prepared samples were boiled for 5 min, run via SDS-PAGE electrophoresis, and transferred to the PVDF membrane (IPVH00010, Millipore, Cork, Ireland) for analysis.

#### Western Blot Antibodies

-ADAM17 (sc-390859, Santa Cruz Biotechnology, Santa Cruz, CA, USA)-γ-secretase antibody sampler kit (#5887, Cell Signaling Technology, Danvers, MA, USA; includes nicastrin, PEN2, Presenilin 1, Presenilin 2)-N-Mertk (AF891, R&D Systems, Minneapolis, MN, USA)-C-Mertk (D21F11, Cell Signaling Technology, Danvers, MA, USA)-B-actin (8H10D10, Cell Signaling Technology, Danvers, MA, USA)-pMertk (p186-749, Phosphosolutions, Aurora, CO, USA)-Myc (71D10, Cell Signaling Technology, Danvers, MA, USA)-Gas6 (SC-376087, Santa Cruz Biotechnology, Santa Cruz, CA, USA)-FLAG-HRP (637311, Biolegend, San Diego, CA, USA)

### 4.7. Production of recombinant Gas6

To produce recombinant Gas6, HEK293 cells, cultured with 10% FBS (*v*/*v*) (R&D Systems, Flowery Branch, GA, USA) in DMEM (Mediatech Inc., Manassa, VA, USA), were transfected with LipoD293 (SignaGen SL100668, Ijamsville, MD, USA) and the pSecTaq-hGas6 plasmid (Tsou et al., JBC, 2014) with the Myc-tag removed. After 18 h in transfection media, cells were washed twice and kept in serum-free DMEM for 72 h. For active, γ–carboxylated Gas6 (Gas6-VK), 2 µg/mL of Vitamin K1 (KJECT 004355, Henry Schein, NY, USA) was added to the media during protein production. For inactive, non-γ-carboxylated Gas6 (Gas6-W), 2 µM of warfarin (81-81-2, Sigma Aldrich) was added to the media during protein production. After transfection, conditioned media were filtered and concentrated using a 10 kDa molecular weight cut-off. After concentrating, serum free DMEM was added to exchange the HEK-conditioned media for fresh media, and it was once again concentrated. Concentrated Gas6 reagents were analyzed for concentration via Western blotting using a Gas6 standard (R&D Systems, 885-GSB-050). Both Gas6-VK and Gas6-W were then diluted with SF DMEM to a stock concentration of 100 nM, with a final use concentration of 10 nM, unless otherwise noted. All steps were followed for the mock transfection control except for the addition of plasmid, Vitamin K, or Warfarin, which was diluted following the Gas6-VK results.

### 4.8. Gas6/Phosphatidylserine (PS) Treatments

To activate Mertk, cells were resuspended in serum-free RPMI media containing up to 10 µM of Gas6-VK or Gas6-W, with concentrations determined via Western blotting. After 30 min, cells were lysed according to the Western blot protocol above. To demonstrate the reduction of the MTag construct with Gas6, cells were resuspended in serumfree RPMI media. An amount of >10 nM of Gas6-VK was added in combination with 1 µM of PS and incubated for up to 6 h.

### 4.9. Flow Cytometry

Flow cytometry analysis was performed using an LSRII (BD Biosciences, Mississauga, Canada). Briefly, THP-1 cells expressing the Mertk-tagged constructs were evaluated with FLAG-PE or Mertk-APC, both recognizing the extracellular domain of the construct. The presence of the C-terminus was determined by detecting the GFP tag. After treatment, cells were washed with chilled flow buffer (PBS + 1% BSA). Data analysis was performed using FlowJo (FlowJo v.10, BD, OR, USA).

#### Flow Antibodies

-Flag PE (637310, Biolegend, CA, USA)-hMertk (FAB8912A, R&D Systems, MN, USA)

### 4.10. Confocal Microscopy

THP-1 cells were seeded to poly-L-lysine-covered cover glass chamber slices (Lab-Teck Chambered#1.5, MI, USA). Cells were then fixed with 4% paraformaldehyde in phosphate-buffered saline (PBS) for 20 min at room temperature and stained with DAPI. The fluorescence images were obtained using a confocal A1R HD25 microscope (Nikon, Japan).

### 4.11. Statistical Significance

Unless otherwise noted, experiments were repeated at least three times. Statistical significance and p values are noted in the figure legends. Differences are noted as mean values plus/minus standard error.Statistical analysis was performed using PrismGraphPad, Version 10.1.2 (GraphPad, CA, USA). Software. Graphs were plotted using Prism or Microsoft Excel (WA, USA).

### 4.12. Figures and Illustration

All figures and scientific illustrations presented in this manuscript were created using Adobe Illustrator software (Adobe Inc. (2019). Adobe Illustrator. Retrieved from https://adobe.com/products/illustrator, accessed on 1 February 2024).

## Figures and Tables

**Figure 1 ijms-25-04404-f001:**
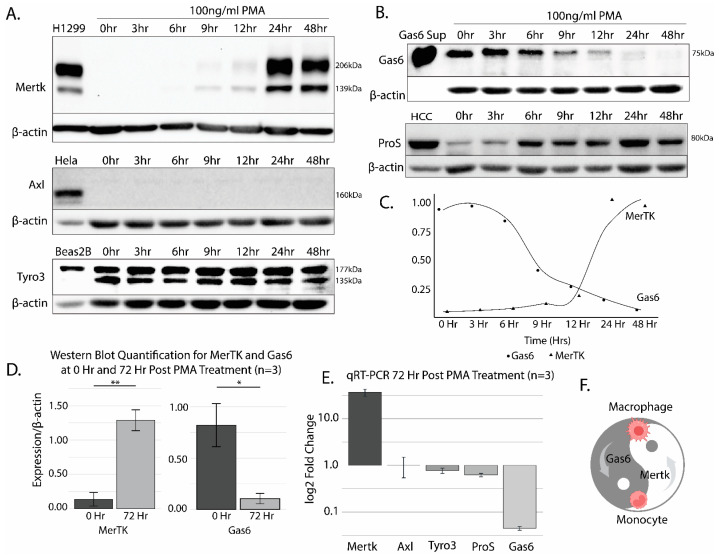
Differentiation of THP-1 cells induced by PMA. (**A**) THP-1 monocytes are differentiated to macrophages and induced to express Mertk by the addition of 100 ng/mL of PMA for at least 24 h, while Axl expression is not induced. Likewise, Tyro3 expression remains stable and is unaffected by treatment. H1299 and Beas2B cells are used as positive controls for Mertk/Axl and Tyro3, respectively. The observed doublet for Mertk corresponds to fully glycosylated and partially glycosylated glycoforms. (**B**) THP-1 cells treated with 100 ng/mL of PMA over 48 h exhibit a decreased expression of Gas6 but an increased expression of Protein S. (**C**) MertK and Gas6 expression patterns over a time of 48 h after 100 ng/mL of PMA treatment as quantified from Western blots. (**D**) Bar plots showing MerTK and Gas6 expression quantified from Western blots at 0 h and post 72 h of 100 ng/mL of PMA treatment (significant differences were observed between the 0 h and 72 h time points for Gas6 (* *p* = 0.029) and MerTK (** *p* = 0.003), as determined by independent *t*-tests). (**E**) THP-1 cells were treated with 100 ng/mL of PMA over 72 h. mRNA was analyzed via qRT-PCR for Mertk, Axl, Protein S (ProS), Tyro3, and Gas6. Mertk transcription was highly elevated, more than 10-fold, due to PMA treatment, while Gas6 transcription was repressed more than 10-fold. (**F**) PMA treatment of THP-1s illustrates a complimentary phenomenon. As a monocyte, Gas6 is expressed with little Mertk expression; however, when differentiated, Mertk is highly expressed in favor of Gas6.

**Figure 2 ijms-25-04404-f002:**
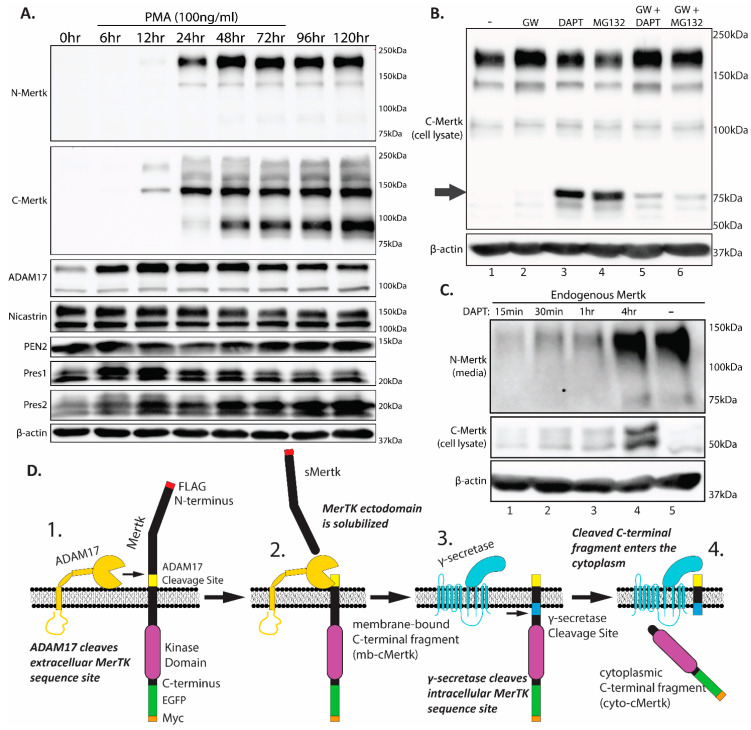
Mertk is sequentially cleaved by ADAM17 and γ-secretase. (**A**) PMA-differentiated THP-1s (100 ng/mL) express Mertk and the necessary cleavage components, ADAM17 and γ-secretase (nicastrin, PEN2, Pres1, and Pres2; APH-1 not shown) Mertk expression confirmed by both N-terminal-specific (N-Mertk) and C-terminal-specific (C-Mertk) antibodies. Note the expression of an unidentified 90 kDa band. An increase in the pro-form (130 kDa) of ADAM17 is observed within 6 h of PMA treatment, though the mature form (100 kDa) remains constant. (**B**) PMA-differentiated THP-1s (for 72 h) were treated for 4 h with either 3 µM of GW280264X (GW; ADAM17 inhibitor), 5 µM of DAPT (γ-secretase inhibitor), or 10 µM of MG132 (proteasomal inhibitor). Both DAPT and MG132 treatment results in a band at approximately 50 kDa, with the MG132 band being slightly lower, suggesting the stabilization of mb-cMertk and cyto-cMertk. The slight difference in molecular weight is attributed to the loss of the transmembrane domain. The addition of GW280264X decreases the presence of both fragments due to the sequential nature of Mertk cleavage. In addition, GW280264X slightly increases full-length Mertk expression, demonstrating the enhanced stability of Mertk. Note the presence of the unidentified 90 kDa band. (**C**) THP-1 cells differentiated with 100 ng/mL of PMA for 72 h were treated with 5 µM of DAPT over the course of 4 h in serum-free media. Stabilization and the subsequent accumulation of the mb-cMertk fragment are observed robustly after 4 h. Lane 5 represents differentiated THP-1 cells after 4 h in untreated serum-free media. (**D**) Mertk is a substrate of ADAM17 and γ-secretase cleavage. Cleavage occurs sequentially. 1. The ADAM17 cleavage site is identified. 2. Active ADAM17 cleaves Mertk, releasing a soluble ectodomain (sMertk) and leaving a membrane-bound C-terminal fragment (mb-cMertk). 3. After ADAM17 cleavage, the γ-secretase complex identifies the remaining membrane-bound fragment of Mertk. 4. γ-secretase cleavage is initiated and releases a C-terminal fragment into the cytosol (cyto-cMertk).

**Figure 3 ijms-25-04404-f003:**
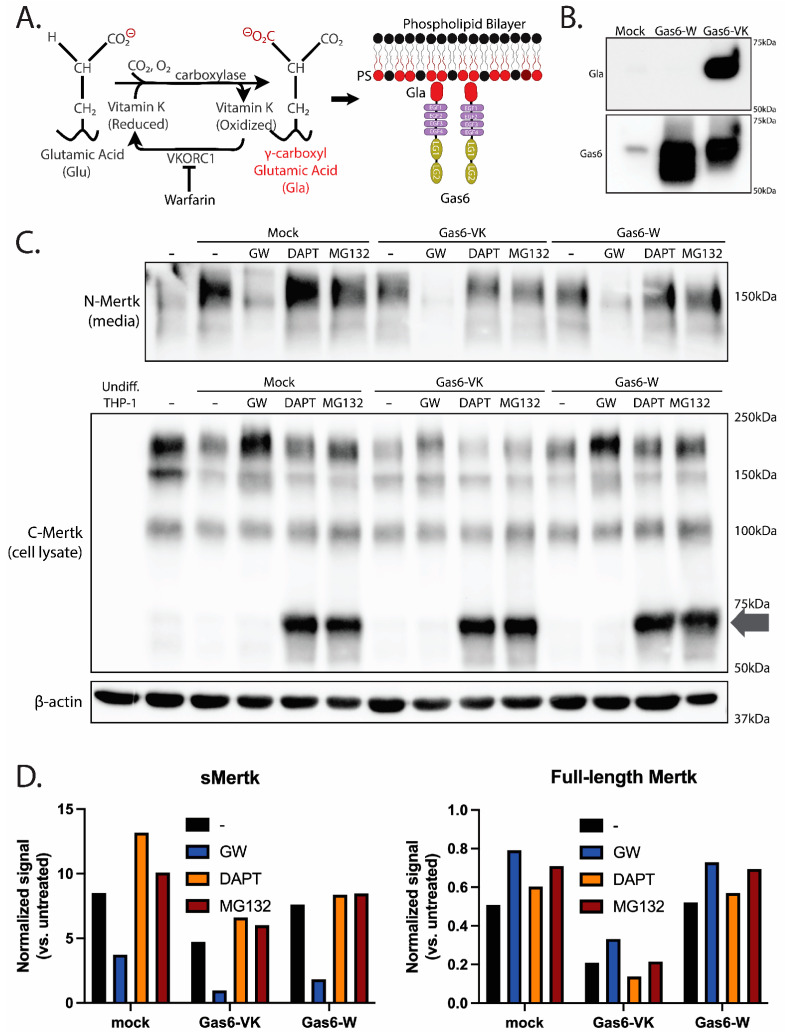
γ-carboxylated Gas6 Reduces Full-length Mertk Expression and Decreases sMertk. (**A**) γ-carboxylation status (Gla) of Gas6 is regulated with the addition of Vitamin K or warfarin, producing either a γ-carboxylated, active ligand or a non-γ-carboxylated, inactive ligand, respectively. The carboxylation status (Gla) domain of Gas6 can bind with externalized PS. (**B**) Recombinant inactive (Gas6-W) and active (Gas6-VK) were produced via HEK293 transfection, with a mock transfection control (Mock). The amount of Gas6 was observed (bottom), while the carboxylation status that is responsible for ligand activity was determined for each (**left**, **top**). (**C**) PMA-differentiated THP-1s were treated for 4 h with either serum-free RPMI only (-), a mock transfection control, 10 nM of active Gas6 (Gas6-VK), or 10 nM of inactive Gas6 (Gas6-W). Treatments were alone or combined with inhibitors of 3 µM of GW280264X (an ADAM17 inhibitor), 5 µM of DAPT (a γ-secretase inhibitor), or 10 µM of MG132 (a proteasomal inhibitor). (**D**) Quantitative results indicate that Gas6, either active or inactive, does not stabilize the C-terminal fragments as shown in the DAPT and MG132 treatments (bands at 75 kDa). As denoted by sMertk (**top**), the cleavage is decreased with GW280264X treatment compared to the untreated control.

**Figure 4 ijms-25-04404-f004:**
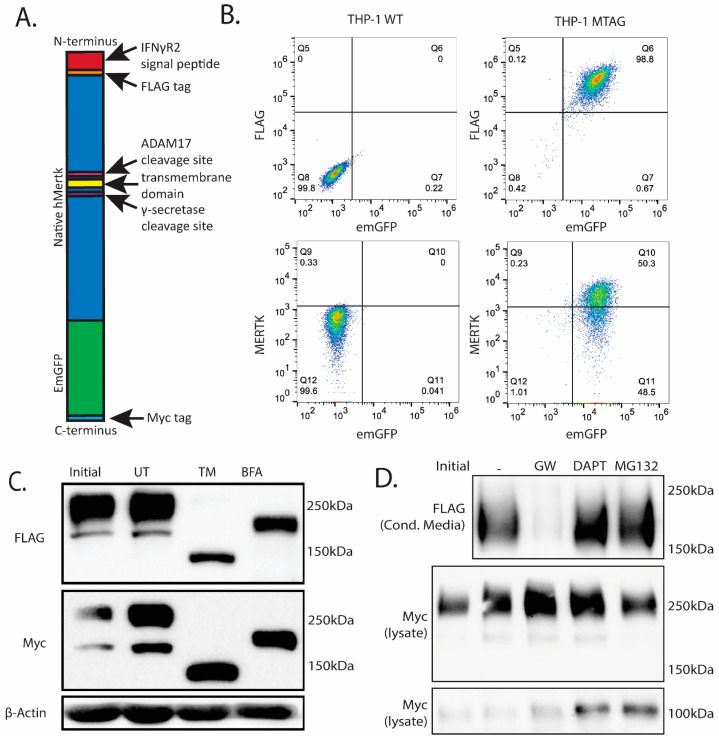
Expression and Validation of tagged Mertk construct in THP-1 cells. (**A**) A diagram of the tagged Mertk construct is shown to scale to illustrate the size of the FLAG, EmGFP, and Myc tags compared to the Mertk receptor. These tags were added to track both the N-terminal and C-terminal domains post-cleavage. Shown are the IFNγR2 signal peptide (red), FLAG-tag (orange), extracellular N-terminal Mertk region (blue), ADAM17 cleavage site (yellow), transmembrane domain (pink), γ-secretase cleavage site (purple), intracellular C-terminal Mertk region (purple), EmGFP tag (green), and Myc tag (orange). (**B**) The expression of the tagged construct was assessed via flow cytometry using anti-FLAG-PE (left) and anti-N-Mertk-APC (**right**). Both signals positively correlate with the intracellular EmGFP signal. (**C**) Expression of the tagged construct was assessed via Western blotting. Cells were treated for 24 h with either tunicamycin (TM) or Brefeldin A (BFA). The untreated (UT) sample was kept in serum-free RPMI media for the duration of the treatment. TM prevents complete glycosylation and results in a 136 kDa band corresponding to the theoretical molecular weight of the construct, while BFA only partially inhibits glycosylation. (**D**) The cleavage of the tagged construct is identical to the native Mertk (from Figure 2).

**Figure 5 ijms-25-04404-f005:**
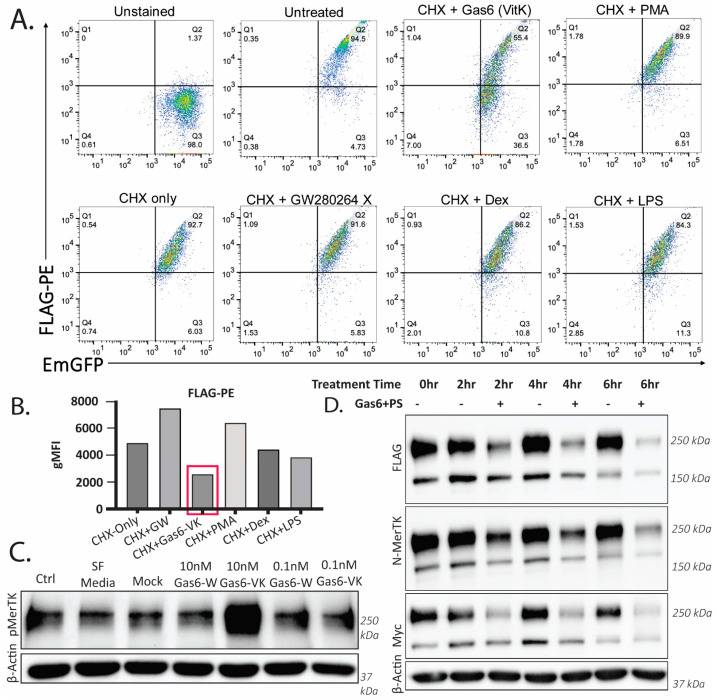
γ-carboxylated Gas6 reduces the tagged Mertk construct on cell membranes of THP-1 cells. (**A**) THP-1s expressing the tagged construct were starved for 18 h in serum-free RPMI and treated for 3 h. Flow cytometry data shows positive GFPs without staining. As expected, CHX treatment after 3 h reduces the expression of the construct. Gas6-VK, used at a concentration > 10 nM, decreases the FLAG-PE signal more when compared to other treatments known to induce cleavage (PMA, LPS). GW280264X is used as a control for Mertk cleavage. (**B**) Histogram analysis of “A.” Gas6-VK induces a shift in the FLAG-PE signal, showing that less surface Mertk is present. γ-Carboxylated Gas6 induced the degradation of Mertk on the cell membrane. (**C**) THP-1s treated with γ-Carboxylated Gas6 at a 10 nM concentration show increased MerTK phosphorylation, detected by immunoblotting against pMerTK. (**D**) THP-1s expressing the tagged construct, treated with >10 nM of Gas6-VK + 1 µM of PS, show a reduced level of both domains of the tagged construct, suggesting that the Gas6-VK + PS treatment is leading to a degradation of the full-length receptor.

**Figure 6 ijms-25-04404-f006:**
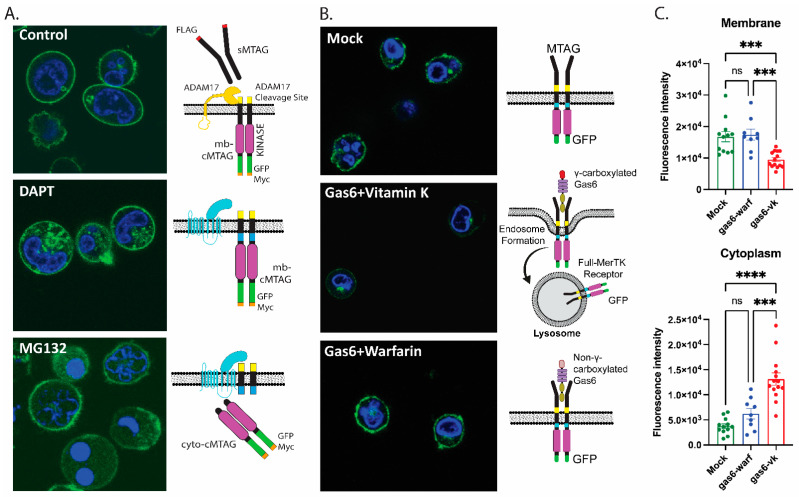
Confocal imaging of the GFP-tagged MerTK construct displayed the differential localization of MerTK upon ligand stimulation and the inhibition of proteases. (**A**) DAPT and MG132 treatments induce increased cytoplasmic GFP signals compared to untreated cells. (**B**) Confocal imaging shows GFP localized on the cell membrane, indicating the presence of tagged Mertk constructs on the cell surface. γ-Carboxylated Gas6-treated tagged THP-1 cells showed a reduction in the MerTK construct from the membrane and the localization in lysosomes. (**C**) Quantification of the GFP fluorescence intensity per cell, calculated from confocal images in mock, γ-carboxylated Gas6, and non-γ-carboxylated Gas6, with the bar plots showing the mean and standard error of each treatment. ns; non-significant; *** *p* < 0.001; **** *p* < 0.0001.

**Figure 7 ijms-25-04404-f007:**
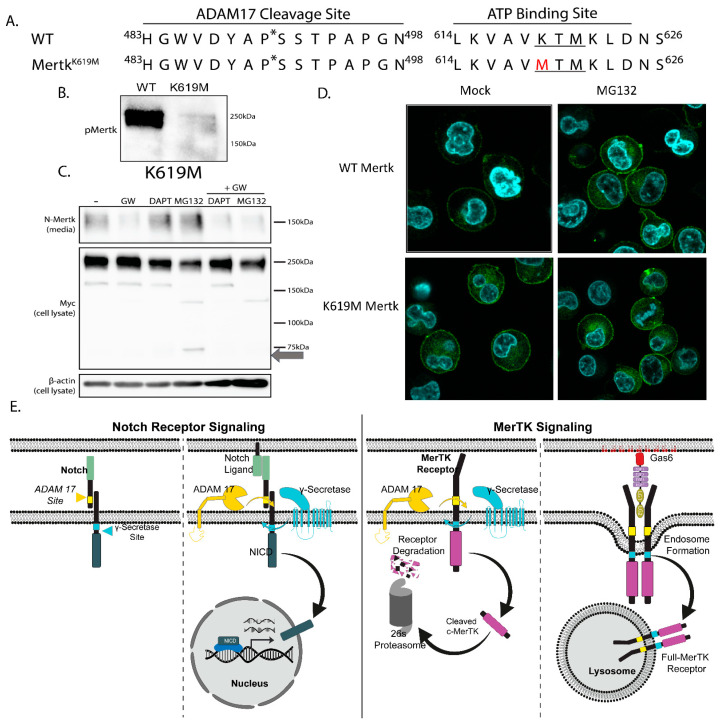
γ-carboxylation of Gas6 induces Mertk degradation independent of phosphorylation. (**A**) Mutants of the tagged construct were created. K619M, a substitution in the ATP-binding site (underlined) of the Mertk kinase domain, inhibits autophosphorylation by preventing ATP binding and the exchange of phosphate molecules. * Amino acid position for ADAM17 cleavage. (**B**) Constructs are treated with 10 nM ofGas6-VK for 30 min after a 6 h serum starvation. With the ability to bind ATP and phosphorylate, the WT construct becomes phosphorylated while the kinase dead K619M mutant does not. (**C**) Mutant constructs were treated with 3 µM of GW280264X (an ADAM17 inhibitor; GW), 5 µM of DAPT (a γ-secretase inhibitor), or 10 µM of MG132 (a proteasomal inhibitor) for 4 h. Results show that the absence of the K619M C-terminal fragment from the MG132 treatment with the addition of GW indicates the fragment is a product of cleavage. (**D**) Confocal images indicated that the K619M mutants have more cytoplasmatic c-Mertk than WT Mertk with MG132 treatment. (**E**) Comparison of the contrasting pathways between Notch Receptor and MerTK Signaling Models. In the absence of a ligand, Notch is not cleaved, while MerTK undergoes homeostatic cleavage, leading to proteasomal degradation. Upon ligand binding, Notch is cleaved at the ADAM 17 site, revealing the gamma-secretase site. Subsequent gamma-secretase cleavage releases the Notch intracellular domain, translocating it to the nucleus for transcriptional activation. Conversely, MerTK, upon ligand (Gas6) interaction, is internalized into endosomal compartments and localizes within lysosomes.

**Table 1 ijms-25-04404-t001:** Sequence of primers for qRT-PCR.

Gene	Forward Primer 5′ -> 3′	Reverse Primer 5′ -> 3′
hMerTK	CAGGAAGATGGGACCTCTCTG	GGCTGAAGTCTTTCATGCACGC
hAxl	GTTTGGAGCTGTGATGGAAGGC	CGCTTCACTCAGGAAATCCTCC
hTyro3	CACTGAGCTGGCTGACTAAGCCCC	AATGCATGCACTTAAGCAGCAGGG
hGas6	GCCTTCTACAGCCTGGACTAC	TCTTGAGTTTCTTCGTGGAGTG
hProS	CCATTCCAGACCAGTGTAG	GGTAACTTCCAGGTGTATTATC

## Data Availability

The data used to support the results of this study are available from the corresponding authors upon request.

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
