# Peer review of "Regulation of Mertk Surface Expression via ADAM17 and γ-Secretase Proteolytic Processing"

_ijms, 2024, doi:10.3390/ijms25084404_

Round 1

Reviewer 1 Report

Comments and Suggestions for Authors

The authors have presented an interesting research work entitled "Homeostatic Regulation of Mertk Expression by ADAM17 and 2-Secretase Proteolytic Processing". This study has suggested that regulated proteolysis of Mertk is tightly controlled to negatively regulate surface receptor levels. However, there are several suggestions that need to be incorporated in the revised manuscript. The manuscript can only be published after incorporating these suggestions:

1. The title is not suitable for the designed research work. It lacks the theme and readability. Kindly rewrite it again.

2. The abstract is too long and descriptive. In my view, it should decipher the summary of the study in a more scientific and concise manner. Hence, the author should rewrite the abstract and focus only on the study's major findings.

3. Spacing is highly inappropriate in the whole manuscript. Kindly check it thoroughly.

4. Kindly provide the implication of this statement: "These data suggest that regulated proteolysis of Mertk is tightly controlled to regulate surface receptor levels negatively" in the introduction section in order to establish the implication of your research study.

5. The quality of Figure 1 is not clear, and it is difficult to see the blots clearly. Kindly provide a clear image with high resolution.

6. What is the statistical significance of the results obtained in this research study?

7. Kindly redraw Figure 2D as the mechanism is not well explained. Also, improve the quality of Figure 2. It is not at all clear.

8. Data significance is not explained in the graph of Figure 3. I would again recommend the potential author improve the quality of graphs and images and increase their resolution for better visibility.

9. I would again recommend the potential author improve the quality of Figure 4 and increase their resolution for better visibility.

10. In Figure 6 E, it seems that there is a double image. words are not at all clear. The whole image is blurred, and the mechanism is not well elaborated.

11. To support your methodology, Kindly add references in the material section. 

12. Kindly add more recent references in order to support your research.

Comments on the Quality of English Language

Spacing and grammatical errors need to be checked thoroughly in the whole manuscript.

Reviewer 2 Report

Comments and Suggestions for Authors

The manuscript by Lahey et al entitled “Homeostatic Regulation of Mertk Expression by ADAM17 and γ-Secretase Proteolytic Processing” sought to unravel how Mertk is post-translationally regulated by cleavage. This is an interesting topic as how Mertk is regulated is important for many fields, including cancer and immunology, as noted by the authors. However, in its current form there are significant issues with respect to this communication that would need to be addressed. There are important flaws in data analysis and experimental design, and additionally care needs to be taken improve the quality of the figures and to correct some apparent mistakes (detailed below).

In essence, the paper posits that there are essential differences in the way Mertk can be processed. In particular, building on prior published results, it is concluded that after ligand engagement, in the absence of enzymatic cleavage, there will be signal transduction but that homeostatic mechanisms will be invoked mediated through receptor internalisation and lysosomal targeting. On the other hand, ADAM17 activity will lead to generation of cleaved soluble Mertk which has been previously shown to exert a dominant negative effect. Gamma secretase activity would be predicted to produce a C-terminal species with a priori prediction of capacity for nuclear localisation. However, γ-secretase or proteosome inhibition led to stabilisation of a truncated form (validated to be a Mertk cleavage product via C-terminal Myc conjugation). This leads to the inevitable conclusion that the impact of the paper would have been improved by showing the functional outcomes of the Mertk regulation by proteolytic processing.

Major issues

1.      In many instances there is no indication of either the biological or technical N, recognising that on some occasions two replicates are shown in the westerns. This is not sufficient in order to be confident of the reproducibility or the fidelity of these results.

2.      The authors extensively use western blots to assess expression of Mertk and its cleavage products but have not shown quantification of the westerns in many instances. The westerns should be uniformly quantified via densitometry and normalised to the loading control. It is difficult to interpret a significant proportion of the results without this. Additionally, some of the westerns have been cropped so close that the bands are cut off. These figures should be regenerated to prevent this, showing more of the surrounding membrane and enabling assessment of the full spread of molecular species generated in each condition.

3.      The Gas6 was not purified (e.g. by HPLC), only “concentrated using a 10kDa molecular weight cut-off.” Thus, there will be many other proteins in the “Gas6” even from cells grown under serum free conditions. This will complicate interpretation.

4.      In figure 3 C (cell lysate membrane) there appears to be a large difference between lane 2 (serum-free RPMI only) and lane 3 (mock transfected treatment), although it does not appear lane 2 have been quantified in Fig3D. The large difference between lane 2 and 3 calls into question the conclusions drawn from this figure, and indeed, from all experiments using the in-house produced Gas6. Given how the Gas6 was produced (see above) it seems highly probable that other proteins are present which may be affecting the cells.

Additional issues:

1.      The authors imply at the start of the results the PMA treatment induces an “M2” like phenotype based on Mertk expression. In the literature PMA treatment of THP-1 cells has been most commonly reported to induce an inactivated “M0” macrophage (Tedesco et al, Front. Pharmacol., 2018 and Genin et al, BMC Cancer, 2015). Assessment of other markers, such as Arg-1 and CD206 should be performed to confirm that the cells are in an “M2” like state instead of relying solely on the Mertk expression.

2.      The figures are very blurry and should be re-exported at a higher quality. Pixilation and small size of the confocal images makes them difficult to interpret. In addition, the labels in the figures are often very difficult to read. In particular, the labels on the flow cytometry plots need to be much clearer, and ideally the axis should be labelled with what it is showing not just the fluorophore. A particular example is Fig4B where the x-axis is labelled “FITC” but is actually intracellular EmGFP signal. While these do have very similar excitation and emission, and so can be detected in the same channels, they are not the same thing.

3.      Methods are incomplete. For example, how were the GFP positive cells sorted? Was this by FACS? The LSRII is mentioned in later sections, but this is an analyser not a sorter.

4.      Unless supplementary information has been missed the primer sequences for the qPCR have not been provided.

5.      The introduction could do with more focus on Mertk, rather than on the roles of the other TAM receptors. For example, specific examples were given for the role of AXL in cancer, I would suggest the reader would benefit more from examples of the role of Mertk in cancer.

Comments on the Quality of English Language

1.      While the language is, overall, appropriate there are significant grammatical and some factual errors. The manuscript would benefit from a detailed copy edit. Selected examples include:

a.       Sometime “γ” is used, sometimes “g”

b.      Line-313—“Having both biologically active versus biologically inactive species of Mertk, we next” – Should ‘Mertk’ be ‘Gas6’ in this sentence?

Round 2

Reviewer 1 Report

Comments and Suggestions for Authors

All the comments have been sorted by the authors, however I found that there is figure 7 is missing in the revised manuscript. Author must incorporate the suggested incorporation. After that manuscript can be considered for publication.

Comments on the Quality of English Language

Check all the grammatical issues throughout the manuscript.
